# Regularization by Denoising Diffusion Process for MRI Reconstruction

**Batu Ozturkler**[1]                                                                        OZT@STANFORD.EDU
[1] *Stanford University*
**Morteza Mardani**[2]                                                                    MMARDANI@NVIDIA.COM
[2] *NVIDIA*
**Arash Vahdat**[2]                                                                        AVAHDAT@NVIDIA.COM
**Jan Kautz**[2]                                                                              JKAUTZ@NVIDIA.COM
**John Pauly**[1]                                                                            PAULY@STANFORD.EDU

**Editors:** Accepted for publication at MIDL 2023

## Abstract

Diffusion models have recently delivered state-of-the-art performance for MRI reconstruction with improved robustness. However, these models still fail when there is a large distribution shift, and their long inference times impede their clinical utility. In this paper, we present regularization by denoising diffusion processes for MRI reconstruction (RED-diff). RED-diff formulates sampling as stochastic optimization, and outperforms diffusion baselines in PSNR/SSIM with $3\times$ faster inference while using the same amount of memory.
**Keywords:** Diffusion models, Regularization by denoising (RED), MRI reconstruction.

## 1. Introduction

Magnetic Resonance Imaging (MRI) is a widely used non-invasive imaging technique due to its ability to generate high-quality images, but acquiring clinical MRI data requires long scan times. Imaging can be sped up by using multiple receiver coils, and by reducing the amount of captured data with Fourier domain (k-space) undersampling (Lustig et al., 2007; Pruessmann et al., 1999). Generative diffusion models gained popularity for MRI reconstruction due to their high sample quality, improving robustness over unrolled methods under distribution shifts (Chung and Ye, 2021; Jalal et al., 2021; Song et al., 2023). Diffusion models can be pretrained for MRI to serve as the data prior and the pretrained model can be used in a plug-and-play fashion by incorporating the forward model at inference time for universally solving downstream reconstruction tasks without the need for re-training or fine-tuning. However, diffusion models still fail dramatically under large distribution shifts such as scan parameter change, or anatomy change between training and testing. Furthermore, inference time for diffusion models is much larger than end-to-end approaches due to the sequential denoising procedure during reverse diffusion, impeding their clinical utility.

Recently, (Mardani et al., 2023) proposed regularization by denoising diffusion (RED-diff) for solving generic inverse problems. RED-diff uses a variational sampler based on a measurement consistency loss and a score matching regularization. In this paper, for the first time, we propose RED-diff for MRI reconstruction. We evaluate RED-diff for MRI reconstruction on FastMRI and Mridata, and show that it achieves state-of-the-art performance across different acceleration rates and anatomies.

---

**Algorithm 1** RED-diff: regularization by denoising diffusion process for MRI

---

**Input:** k-space data $y$; acquisition model $A = \Omega F S$; $\{\alpha_t, \sigma_t, \lambda_t\}_{t=1}^{T}$
**Initialize:** $\mu = x_{zf} = A^{-1}y$
1: **for** $t = T, ..., 1$ **do**
2:    $\epsilon \sim \mathcal{N}(0, I)$
3:    $x_t = \alpha_t \mu + \sigma_t \epsilon$
4:    $loss = \|A\mu - y\|^2 + \lambda_t (\text{sg}[\epsilon_\theta(x_t; t) - \epsilon])^T \mu$
5:    $\mu \leftarrow \text{OptimizerStep}(loss)$
6: **end for**
7: **return** $\mu$

---

## 2. Methods

**Accelerated MRI.** The forward model for accelerated MRI is given by $y = \Omega F S x + \nu$ where $y$ is the measurement, $x$ is the real image, $S$ are sensitivity maps, $F$ is the Fourier transform, $\Omega$ is the subsampling mask, $\nu$ is noise, and $A = \Omega F S$ is the forward model.

**Diffusion models.** Diffusion models consist of two processes: a forward process that gradually adds noise to input images and a reverse process that learns to generate images by iterative denoising. A popular class of diffusion models uses the variance preserving stochastic differential equation (VP-SDE) (Song et al., 2020). The forward and reverse process is characterized by the noise schedule $\beta(t)$ with $t \in [0, T]$ where $t$ is the timestep. $\beta(t)$ is designed such that the final distribution of $x_T$ at the end of the process converges to a standard Gaussian distribution. The reverse generative process requires estimating the score function $\nabla_{x_t} \log p(x_t)$, which denotes the score function of diffused data at time t. $\nabla_{x_t} \log p(x_t)$ can be estimated by training a joint neural network, denoted as $\epsilon_\theta(x_t; t)$, via denoising score matching (Vincent, 2011). For denoising score matching, diffused samples are generated by $x_t = \alpha_t x_0 + \sigma_t \epsilon$ where $\epsilon \sim \mathcal{N}(0, I)$ $x_0 \sim p_{\text{data}}$ is the data distribution, $\sigma_t = 1 - e^{-\int_0^t \beta(s)ds}$, and $\alpha_t = \sqrt{1 - \sigma_t^2}$, and $\epsilon_\theta(x_t; t) \approx -\sigma_t \nabla_{x_t} \log p(x_t)$.

**RED-diff.** (Mardani et al., 2023) proposes a variational inference approach based on KL minimization that corresponds to minimizing a measurement consistency loss equipped with a score-matching regularization term imposed by the diffusion prior. For MRI reconstruction, we consider the following minimization problem

$$\min_\mu \|A\mu - y\|^2 + \mathbb{E}_{t,\epsilon}[w(t)\|\epsilon_\theta(x_t; t) - \epsilon\|_2^2] \tag{1}$$

where $x_t = \alpha_t \mu + \sigma_t \epsilon$, and $w(t)$ is a time-dependent weighting mechanism. To search for $\mu$, we use first-order stochastic optimization. We define the loss per timestep based on the instantaneous gradient by detaching it at each timestep. Then, we can form the loss at time step t as $\|A\mu - y\|^2 + \lambda_t (\text{sg}[\epsilon_\theta(x_t; t) - \epsilon])^T \mu$ where $\lambda_t$ is the weighting term, and sg denotes stopped-gradient, indicating that score is not differentiated during the optimization. We set $\lambda_t = \lambda \sigma_t / \alpha_t$, where $\lambda$ is a hyperparameter. Our full method is described in Alg. 1.

## 3. Results and Discussion

We use the multi-coil fastMRI brain dataset (Zbontar et al., 2018) with 1D equispaced undersampling, and the fully-sampled 3D fast-spin echo multi-coil knee MRI dataset from

| Anatomy | Brain | Knee | | Timing |
| --- | --- | --- | --- | --- |
| $R$ | $R = 4$ | $R = 12$ | $R = 16$ | (sec/iter) |
| Zero-filled | 27.8/0.81 | 24.5/0.63 | 24.0/0.60 | - |
| CSGM-Langevin | 36.3/0.78 | 31.4/**0.82** | 31.8/**0.79** | 0.344 |
| RED-diff | **37.1/0.83** | **33.2**/0.78 | **32.7**/0.77 | **0.114** |

Table 1: Reconstruction PSNR/SSIM for fastMRI brain and Mridata knee dataset.

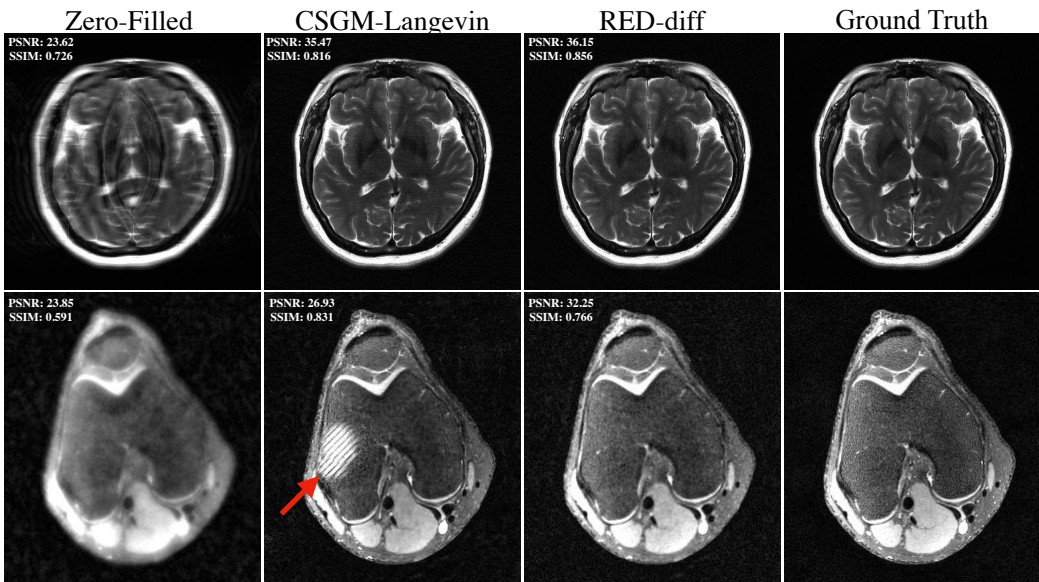

Figure 1: Example reconstruction for brain at $R = 4$, and knee at $R = 12$.

(Ong et al., 2018) with 2D Poisson Disc undersampling mask, as in (Jalal et al., 2021). We used 6 validation volumes for fastMRI, and 3 volumes for Mridata by selecting 32 middle slices from each volume. Both datasets had a total of 96 test slices. For RED-diff, we use linear schedule for $\beta(t)$ from 0.0001 to 0.02, and $T = 1000$. We adopt Adam optimizer with initial learning rate 0.1 and no weight decay regularization, and set the momentum to $(0.9, 0.99)$ where $\lambda = 0.25$. We compare RED-diff with CSGM-Langevin (Jalal et al., 2021). For CSGM-Langevin and RED-diff, we use the score function from (Jalal et al., 2021) which was trained on a subset of the FastMRI multi-coil brain dataset. We evaluate the methods in i) the in-distribution setting on brain at $R = 4$, ii) the out-of-distribution setting with knee at $R = \{12, 16\}$. Table 1 shows comparison of reconstruction methods for FastMRI brain, and Mridata knee datasets. RED-diff outperforms CSGM-Langevin in most cases, with a PSNR improvement of +0.7dB for brain, +1.8dB for knee, and an SSIM improvement of +0.05 for brain, while having $3\times$ faster inference time using same amount of memory. Fig.1 shows example reconstructions for brain at $R = 4$, and knee at $R = 12$. RED-diff produces higher quality reconstruction in both cases. Crucially, it is observed that CSGM-Langevin is sensitive in the out-of-distribution setting and produces hallucination artifacts, whereas RED-diff mitigates these artifacts and produces a reconstruction with no hallucinations. In conclusion, RED-diff improves reconstruction quality for MRI reconstruction, and speeds up inference by at least $3\times$ while using the same inference memory.

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
