# OpenReview forum: "Regularization by Denoising Diffusion Process for MRI Reconstruction"
_MIDL.io/2023/Short_Paper_Track — MIDL 2023 Short paper track Poster_

### Official Review · Reviewer_R5tn · 2023-04-24
**The authors presented RED-diff, a regularization by denoising diffusion process for MRI reconstruction. The RED-diff algorithm formulated the sampling process as a stochastic optimization and minimized the inference time while improving PSNR/SSIM of the existing diffusion models.**

**Rating:** 9
**Confidence:** 5

**Review:**

Overall, the technical contribution of this paper is well-presented; hence is easy for readers to follow. This reviewer does not have major concerns but a few minor comments.

Strengths:

-Integration of the Fourier transform (with the forward process of the diffusion model) in the variational inference approach to speed up the experiment process.

-Clear derivation and description of the forward and backward processes and its integration to the loss function.

-Clear discussion and justification of the experimental settings and outcomes.

Weaknesses:

-The authors did not mention, discuss, or provide reference(s) about the ‘Zero-filled’ baseline approach which makes it unclear about the comparative analysis.

-Experimental baselines and comparative analysis: the authors need to consider some of the existing models (i.e., [1-3]) for performance comparison to show the efficiency of the proposed model (as the SSIM in Table 1 does not seem to achieving the optimal performance).

[1] Chung, H., Lee, E. S., & Ye, J. C. (2022). MR Image Denoising and Super-Resolution Using Regularized Reverse Diffusion. IEEE Transactions on Medical Imaging.

[2] Chung, H., & Ye, J. C. (2022). Score-based diffusion models for accelerated MRI. Medical Image Analysis, 80, 102479.

[3] Peng, C., Guo, P., Zhou, S. K., Patel, V. M., & Chellappa, R. (2022, September). Towards performant and reliable undersampled MR reconstruction via diffusion model sampling. In Medical Image Computing and Computer Assisted Intervention–MICCAI 2022: 25th International Conference, Singapore, September 18–22, 2022, Proceedings, Part VI (pp. 623-633). Cham: Springer Nature Switzerland.

---

### Official Review · Reviewer_8gJv · 2023-04-24
**Regularization by Denoising Diffusion Process for MRI Reconstruction**

**Rating:** 9
**Confidence:** 4

**Review:**

This work proposes a new method called RED-diff, which utilizes diffusion models to solve MRI reconstruction tasks with improved performance, faster inference times and better robustness under distribution shifts. The authors argue that while diffusion models have shown significant promise in the field, they still fail when faced with large distribution shifts and their long inference times make them impractical for clinical use.

The paper builds upon previous work by utilizing regularization by denoising diffusion (RED) to solve inverse problems in MRI. Specifically, the authors extend RED to MRI reconstruction by formulating sampling as stochastic optimization. The proposed approach is evaluated on FastMRI and Mridata datasets and achieves state-of-the-art performance across different acceleration rates and anatomies.

Overall, the paper is well-written and clearly presents the motivation, methodology, and results. The use of diffusion models and regularization by denoising is an innovative approach that has shown to be effective for MRI reconstruction. The authors provide a thorough explanation of the proposed method, including the mathematical formalism and algorithmic details. The experiments are carefully designed and the results demonstrate the superior performance of the proposed approach compared to diffusion baselines.

Some potential limitations of the paper include the lack of a detailed comparison with other state-of-the-art methods and the relatively limited discussion of the significance of the results. Additionally, the paper does not provide a detailed analysis of the sensitivity of the proposed method to hyperparameters. Nevertheless, the paper represents an important contribution to the field of MRI reconstruction and diffusion models, and has the potential to have a significant impact on clinical applications of MRI.